# Pathophysiology and Treatment of Stroke: Present Status and Future Perspectives

**DOI:** 10.3390/ijms21207609

**Published:** 2020-10-15

**Authors:** Diji Kuriakose, Zhicheng Xiao

**Affiliations:** Development and Stem Cells Program, Monash Biomedicine Discovery Institute and Department of Anatomy and Developmental Biology, Monash University, Melbourne, VIC 3800, Australia; diji.kuriakose@monash.edu

**Keywords:** stroke, pathophysiology, treatment, neurological deficit, recovery, rehabilitation

## Abstract

Stroke is the second leading cause of death and a major contributor to disability worldwide. The prevalence of stroke is highest in developing countries, with ischemic stroke being the most common type. Considerable progress has been made in our understanding of the pathophysiology of stroke and the underlying mechanisms leading to ischemic insult. Stroke therapy primarily focuses on restoring blood flow to the brain and treating stroke-induced neurological damage. Lack of success in recent clinical trials has led to significant refinement of animal models, focus-driven study design and use of new technologies in stroke research. Simultaneously, despite progress in stroke management, post-stroke care exerts a substantial impact on families, the healthcare system and the economy. Improvements in pre-clinical and clinical care are likely to underpin successful stroke treatment, recovery, rehabilitation and prevention. In this review, we focus on the pathophysiology of stroke, major advances in the identification of therapeutic targets and recent trends in stroke research.

## 1. Introduction

Stroke is a neurological disorder characterized by blockage of blood vessels. Clots form in the brain and interrupt blood flow, clogging arteries and causing blood vessels to break, leading to bleeding. Rupture of the arteries leading to the brain during stroke results in the sudden death of brain cells owing to a lack of oxygen. Stroke can also lead to depression and dementia.

Until the International Classification of Disease 11 (ICD-11) was released in 2018, stroke was classified as a disease of the blood vessels. Under the previous ICD coding rationale, clinical data generated from stroke patients were included as part of the cardiovascular diseases chapter, greatly misrepresenting the severity and specific disease burden of stroke. Due to this misclassification within the ICD, stroke patients and researchers did not benefit from government support or grant funding directed towards neurological disease. After prolonged advocacy from a group of clinicians, the true nature and significance of stroke was acknowledged in the ICD-11; stroke was re-categorized into the neurological chapter [1]. The reclassification of stroke as a neurological disorder has led to more accurate documentation of data and statistical analysis, supporting improvements in acute healthcare and acquisition of research funding for stroke.

## 2. Epidemiology of Stroke

Stroke is the second leading cause of death globally. It affects roughly 13.7 million people and kills around 5.5 million annually. Approximately 87% of strokes are ischemic infarctions, a prevalence which increased substantially between 1990 and 2016, attributed to decreased mortality and improved clinical interventions. Primary (first-time) hemorrhages comprise the majority of strokes, with secondary (second-time) hemorrhages constituting an estimated 10–25% [2,3]. The incidence of stroke doubled in low-and-middle income countries over 1990–2016 but declined by 42% in high-income countries over the same period. According to the Global Burden of Disease Study (GBD), although the prevalence of stroke has decreased, the age of those affected, their sex and their geographic location mean that the socio-economic burden of stroke has increased over time [3].

*Age-specific stroke*: The incidence of stroke increases with age, doubling after the age of 55 years. However, in an alarming trend, strokes in people aged 20–54 years increased from 12.9% to 18.6% of all cases globally between 1990 and 2016. Nevertheless, age-standardized attributable death rates decreased by 36.2% over the same period [3,4,5]. The highest reported stroke incidence is in China, where it affects an estimated 331–378 individuals per 100,000 life years. The second-highest rate is in eastern Europe (181–218 per 100,000 life years) and the lowest in Latin America (85–100 per 100,000 life years) [3].

*Gender-specific stroke*: The occurrence of stroke in men and women also depends on age. It is higher at younger ages in women, whereas incidence increases slightly with older age in men. The higher risk for stroke in women is due to factors related to pregnancy, such as preeclampsia, contraceptive use and hormonal therapy, as well as migraine with aura. Atrial fibrillation increases stroke risk in women over 75 years by 20%. Based on the National Institutes of Health Stroke Scale (0 = no stroke, 1–4 = minor stroke, 5–15 = moderate stroke, 15–20 = moderate/severe stroke, 21–42 = severe stroke), mean stroke severity was estimated at 10 for women and 8.2 for men. Both brain infarction and intracerebral hemorrhage (ICH) are common in men, but cardioembolic stroke, a more severe form of stroke, is more prevalent among women. The fatality rate for stroke is also higher among women [5,6,7]. Women live longer than men, which is one reason for their higher incidence of stroke; another important concern is women’s delay in accepting help for ongoing symptoms [8]. For men, the most common causes of stroke are tobacco smoking, excessive alcohol consumption, myocardial infarction and arterial disorders [9].

*Geographic and racial variation*: As noted earlier, stroke incidence varies considerably across the globe. A global population-based study of the prevalence of stroke and related risks examined demography, behavior, physical characteristics, medical history and laboratory reports, and revealed the contribution of exposure to air pollution and particulate matter to stroke mortality [10]. Another population-based study, conducted in north-eastern China, is thought to be broadly representative of the disease situation in developing countries. It found hypertension to be a statistically significant risk for stroke, specifically ischemic stroke [11]. A study conducted in the United States (US) also identified hypertension as a major cause of stroke and described geographical variation in symptomatic intensity in stroke sufferers. Insufficient physical activity, poor food habits and nicotine and alcohol consumption were considered added risks [12]. Differences in exposure to environmental pollutants, such as lead and cadmium, also influenced stroke incidences across regions. This study also revealed differences in stroke incidence between non-Hispanic white and black populations aged 40–50 years [13].

*Socioeconomic variation*: There is a strong inverse relationship between stroke and socioeconomic status, attributable to inadequate hospital facilities and post-stroke care among low-income populations [14]. A case study conducted in the US showed that people with high financial status had better stroke treatment options than deprived individuals [15]. A study in China linked low income and lack of health insurance to prevention of secondary stroke attack [16]. Research conducted in Austria associated level of education with take-up of treatments such as echocardiography and speech therapy; however, there was no difference in administration of thrombolysis, occupational therapy, physiotherapy or stroke care for secondary attack by socioeconomic status [17]. Similarly, in the Scottish healthcare system, basic treatments like thrombolysis were provided irrespective of the economic status of patients [18].

## 3. Pathophysiology of Stroke

Stroke is defined as an abrupt neurological outburst caused by impaired perfusion through the blood vessels to the brain. It is important to understand the neurovascular anatomy to study the clinical manifestation of the stroke. The blood flow to the brain is managed by two internal carotids anteriorly and two vertebral arteries posteriorly (the circle of Willis). Ischemic stroke is caused by deficient blood and oxygen supply to the brain; hemorrhagic stroke is caused by bleeding or leaky blood vessels.

Ischemic occlusions contribute to around 85% of casualties in stroke patients, with the remainder due to intracerebral bleeding. Ischemic occlusion generates thrombotic and embolic conditions in the brain [19]. In thrombosis, the blood flow is affected by narrowing of vessels due to atherosclerosis. The build-up of plaque will eventually constrict the vascular chamber and form clots, causing thrombotic stroke. In an embolic stroke, decreased blood flow to the brain region causes an embolism; the blood flow to the brain reduces, causing severe stress and untimely cell death (necrosis). Necrosis is followed by disruption of the plasma membrane, organelle swelling and leaking of cellular contents into extracellular space [20], and loss of neuronal function. Other key events contributing to stroke pathology are inflammation, energy failure, loss of homeostasis, acidosis, increased intracellular calcium levels, excitotoxicity, free radical-mediated toxicity, cytokine-mediated cytotoxicity, complement activation, impairment of the blood–brain barrier, activation of glial cells, oxidative stress and infiltration of leukocytes [21,22,23,24,25].

Hemorrhagic stroke accounts for approximately 10–15% of all strokes and has a high mortality rate. In this condition, stress in the brain tissue and internal injury cause blood vessels to rupture. It produces toxic effects in the vascular system, resulting in infarction [26]. It is classified into intracerebral and subarachnoid hemorrhage. In ICH, blood vessels rupture and cause abnormal accumulation of blood within the brain. The main reasons for ICH are hypertension, disrupted vasculature, excessive use of anticoagulants and thrombolytic agents. In subarachnoid hemorrhage, blood accumulates in the subarachnoid space of the brain due to a head injury or cerebral aneurysm (Figure 1) [27,28].

## 4. Risk Factors for Stroke

As noted earlier, the risk of stroke increases with age and doubles over the age of 55 years in both men and women. Risk is increased further when an individual has an existing medical condition like hypertension, coronary artery disease or hyperlipidemia. Nearly 60% of strokes are in patients with a history of transient ischemic attack (TIA). Some of the risk factors for stroke are modifiable, and some are non-modifiable (Figure 2).

### 4.1. Non-Modifiable Risk Factors

These include age, sex, ethnicity, TIA and hereditary characteristics. In the US in 2005, the average age of incidence of stroke was 69.2 years [2,29,30]. Recent research has indicated that people aged 20–54 years are at increasing risk of stroke, probably due to pre-existing secondary factors [31]. Women are at equal or greater risk of stroke than men, irrespective of age [32]. US research shows that Hispanic and black populations are at higher risk of stroke than white populations; notably, the incidence of hemorrhagic stroke is significantly higher in black people than in age-matched white populations [33,34,35].

Transient ischemic attack is classified as a mini stroke; the underlying mechanism is the same as for full-blown stroke. In TIA, the blood supply to part of the brain is blocked temporarily. It acts as a warning sign before the actual event, providing an opportunity to change lifestyle and commence medications to reduce the chance of stroke [36,37].

Genetics contribute to both modifiable and non-modifiable risk factors for stroke. Genetic risk is proportional to the age, sex and race of the individual [38,39], but a multitude of genetic mechanisms can increase the risk of stroke. Firstly, a parental or family history of stroke increases the chance of an individual developing this neurological disorder. Secondly, a rare single gene mutation can contribute to pathophysiology in which stroke is the primary clinical manifestation, such as in cerebral autosomal dominant arteriopathy. Thirdly, stroke can be one of many after-effects of multiple syndromes caused by genetic mutation, such as sickle cell anemia. Fourthly, some common genetic variants are associated with increased stroke risk, such as genetic polymorphism in 9p21 [40]. A genome-wide association study of stroke showed high heritability (around 40%) for large blood vessel disease, and low heritability (16.7%) for small vessel disorders. Recent evidence suggests that studying heritability will improve the understanding of stroke sub-types, improve patient management and enable earlier and more efficient prognosis [5,41].

### 4.2. Modifiable Risk Factors

These are of paramount importance, because timely and appropriate medical intervention can reduce the risk of stroke in susceptible individuals. The major modifiable risk factors for stroke are hypertension, diabetes, lack of physical exercise, alcohol and drug abuse, cholesterol, diet management and genetics.

*Hypertension*: It is one of the predominant risk factors for stroke. In one study, a blood pressure (BP) of at least 160/90 mmHg and a history of hypertension were considered equally important predispositions for stroke, with 54% of the stroke-affected population having these characteristics [42,43]. BP and prevalence of stroke are correlated in both hypertensive and normal individuals. A study reported that a 5–6 mm Hg reduction in BP lowered the relative risk of stroke by 42% [44]. Randomized trials of interventions to reduce hypertension in people aged 60+ have shown similar results, lowering the incidences of symptoms of stroke by 36% and 42%, respectively [45,46].

*Diabetes*: It doubles the risk of ischemic stroke and confers an approximately 20% higher mortality rate. Moreover, the prognosis for diabetic individuals after a stroke is worse than for non-diabetic patients, including higher rates of severe disability and slower recovery [47,48]. Tight regulation of glycemic levels alone is ineffective; medical intervention plus behavioral modifications could help decrease the severity of stroke for diabetic individuals [49].

*Atrial fibrillation (AF)*: AF is an important risk factor for stroke, increasing risk two- to five-fold depending upon the age of the individual concerned [50]. It contributes to 15% of all strokes and produces more severe disability and higher mortality than non-AF-related strokes [51]. Research has shown that in AF, decreased blood flow in the left atrium causes thrombolysis and embolism in the brain. However, recent studies have contradicted this finding, citing poor evidence of sequential timing of incidence of AF and stroke, and noting that in some patients the occurrence of AF is recorded only after a stroke. In other instances, individuals harboring genetic mutations specific to AF can be affected by stroke long before the onset of AF [52,53]. Therefore, we need better methods of monitoring the heart rhythms that are associated with the vascular risk factors of AF and thromboembolism.

*Hyperlipidemia*: It is a major contributor to coronary heart disease, but its relationship to stroke is complicated. Total cholesterol is associated with risk of stroke, whereas high-density lipoprotein (HDL) decreases stroke incidence [54,55,56]. Therefore, evaluation of lipid profile enables estimation of the risk of stroke. In one study, low levels of HDL (<0.90 mmol/L), high levels of total triglyceride (>2.30 mmol/L) and hypertension were associated with a two-fold increase in the risk of stroke-related death in the population [55].

*Alcohol and drug abuse*: The relationship between stroke risk and alcohol intake follows a curvilinear pattern, with the risk related to the amount of alcohol consumed daily. Low to moderate consumption of alcohol (≤2 standard drinks daily for men and ≤1 for women) reduces stroke risk, whereas high intake increases it. In contrast, even low consumption of alcohol escalates the risk of hemorrhagic stroke [57,58,59]. Regular use of illegitimate substances such as cocaine, heroin, phencyclidine (PCP), lysergic acid diethylamide (LSD), cannabis/marijuana or amphetamines is related to increased risk of all subtypes of strokes [60]. Illicit drug use is a common predisposing factor for stroke among individuals aged below 35 years. US research showed that the proportion of illicit drug users among stroke patients aged 15–44 years was six times higher than among age-matched patients admitted with other serious conditions [61]. However, there is no strong evidence to confirm these findings, and the relationship between these drugs and stroke is anecdotal [62].

*Smoking*: Tobacco smoking is directly linked to increased risk of stroke. An average smoker has twice the chance of suffering from a stroke of a non-smoker. Smoking contributes to 15% of stroke-related mortality. Research suggests that an individual who stops smoking reduces the relative risk of stroke, while prolonged second-hand smoking confers a 30% elevation in the risk of stroke [63,64,65].

*Insufficient physical inactivity and poor diet* are associated with increased risk for stroke. Lack of exercise increases the chances of stroke attack in an individual. Insufficient physical activity is also linked to other health issues like high BP, obesity and diabetes, all conditions related to high stroke incidence [66,67]. Poor diet influences the risk of stroke, contributing to hypertension, hyperlipidemia, obesity and diabetes. Certain dietary components are well known to heighten risk; for example, excessive salt intake is linked to high hypertension and stroke. Conversely, a diet high in fruit and vegetables (notably, the Mediterranean diet) has been shown to decrease the risk of stroke [68,69,70,71,72].

## 5. Animal Models of Stroke

Animal models usually used for research include induced, spontaneous, negative and orphan models. In the induced model, a disease condition is induced in the animal with a view to studying the effects, whereas in the spontaneous model, an animal is selected with a similar disease state naturally present in the model. Negative animal models are used to study the resistance mechanisms underlying a particular disease condition. Orphan models are deployed to understand the pathology of a newly characterized disease in human subjects [73,74].

Many animal models have been developed to study the pathophysiology associated with stroke; they offer several advantages over studying stroke in humans or in vitro. The nature of stroke in humans is unpredictable, with diverse clinical manifestation and localization, whereas animal models are highly predictable and reproducible. Pathophysiological investigation often requires direct access to brain tissue, which is possible with animal models but not in humans. Moreover, current imaging techniques are unable to characterize events occurring within the first few minutes of a stroke. Finally, some aspects of stroke, such as vasculature and perfusion, cannot be studied in in vitro models [75]. Different stroke models used in animals are described in the session below (Table 1).

*The intraluminal suture MCAo model*: The middle cerebral artery (MCA) is vulnerable to ischemic insult and occlusion in humans, accounting for 70% of stroke-related disability. This disease model has been widely studied in rat and mouse models, with more than 2600 experiments conducted [76,77]. The MCAo procedure is minimally invasive; it involves occlusion of the carotid artery by insertion of a suture until it interrupts blood flow to the MCA. This procedure is applied for time periods such as 60 or 90 min or permanently, to induce infarction, and has a success rate of 88–100% in rats and mice [78]. The most commonly used animal for studying pre-clinical stroke is the Sprague–Dawley rat, which has a small infarct volume [79]. In mice, C57BL/6 and SV129 are commonly used to introduce MCA infarction. The reproducibility of the technique depends on a multitude of factors, such as the animal strain, suture diameter, body weight and age. The advantage of this model is that it mimics the human ischemic stroke and displays similar penumbra [80]. The MCAo model is appropriate for reproducing ischemic stroke and associated clinical manifestations such as neuronal cell death, cerebral inflammation and blood–brain barrier damage [75].

*Craniectomy model*: This model uses a surgical procedure for inducing occlusion in the artery. In this technique, a neurological deficit can be induced in mice by electrocoagulation causing permanent insult or a microaneurysm until blood flow is interrupted. Alternatively, three-vessel occlusion is used, reducing the blood flow and resulting in damaged tissue. The infarct volume differs depending on whether the occlusion is permanent or transient [81,82,83]. A study conducted in neonatal P14–P18 rats mimicked pediatric stroke in a younger human population; a 3-h occlusion was performed to induce lesions affecting 40–50% of the brain [84]. Similarly, in P7 rats, oedema formation was observed in the MCA, followed by microglial infiltration. The P12 CB-17 is another animal model used for stroke research, mainly due to low variability in occlusion insult to the brain [85]. The other advantages of this model include reproducible infarct size and neurofunctional deficits, reduced mortality and visual ratification. The CB-17 model was successfully used to reproduce cerebral infarction and long-term survival rate, and to study ischemic reperfusion. Researchers showed that reperfusion supports neuron survival, rescues vascular phenotypes and is associated with functional recovery after stroke [86].

*The Levine–Rice model*: It involves histological examination and behavioral tests in rat pups, and it is used to study neonatal hypoxic-ischemic stroke [87]. In this model, a unilateral ligation is followed by reperfusion and recovery. Later, the animal is placed in a hypoxic chamber to understand neonatal stroke pathophysiology as well as regenerative and rehabilitative therapeutic possibilities. P7 rat animal models are commonly used to study the clinical manifestations of hypoxic-ischemic injury [88,89,90].

*Photo-thrombosis model*: This model is based on photo-oxidation of the vasculature leading to lesion formation in the cortex and striatum. In this method, the skull is irradiated with a photoactive dye that causes endothelial damage, intraparenchymal vessel aggregation and platelet stimulation in the affected area. It is injected intraperitoneally in mice and intravenously in rats [91]. This model is highly reproducible, with a low mortality rate and no surgery. The pathophysiology of this method is slightly different to that seen in human stroke due to little collateral blood flow or formation of ischemic penumbra. However, recent researchers modified the photothrombotic ischemia model to include hypoperfusion in an attempt to mimic penumbra. It has also been deployed in freely moving mice to evaluate the development of motor cortex ischemia and motor deficits. This model permits assessment of the ongoing infarction and improves our understanding of the neuronal insult and repair process [92,93].

*Endothelin-1 model: Endothelin-1 (ET-1)*: ET-1 is a small peptide molecule produced by smooth muscle cells and the endothelium. It is a paracrine factor that restricts the vascular system through cell-specific receptors. Ischemic lesion is induced by stereotaxic injection of ET-1 directly into the exposed MCA in the intracerebral or cortex region [94]. ET-1 administration was observed to cause 70–90% reduction in cerebral blood flow, followed by reperfusion [95]. This technique is minimally invasive, has a low death rate and can be applied to deep and superficial brain regions. It is appropriate for long-term lesion studies, and the lesion size can be controlled by regulating ET-1 concentration, which is critical for reproducibility [95]. ET-1 is expressed by both neurons and astrocytes, which may decrease the stringency of interpretation of neuronal dysfunction in stroke [96]. A study in juvenile P21 rats used ET-1 to induce focal lesion in the striatum [97]. Similarly, aged P12 and P25 rats showed neuronal damage and lesion formation after injection of ET-1 into the hippocampus [98].

*The embolic stroke model*: It includes microsphere, macrosphere and thromboembolic models. The microsphere model involves introduction of spheres of diameter 20–50 μm into the circulatory system using a microcatheter to form multifocal infarcts [99]. Macrospheres are 100–400 μm in diameter and introduced into the intracerebral artery (ICA) to produce reproducible lesions in the MCA [100]. In the thromboembolic model, thrombin is directly injected to form clots in the ICA or MCA. The volume of the infarct depends upon the size of the clot formed [101]. This model closely resembles the type of stroke seen in humans. Prior study of clots induced by this model in mice have showed that they are mainly comprised of polymerized fibrin with few cells and platelets present, and 75% of clots exhibit platelet/fibrin build-up and deposition of neutrophils, monocytes and erythrocytes [102].

*Neurorehabilitation in animal models*: Various rehabilitative devices and forced training strategies have been deployed in stroke-affected animals to study neurological behavior. Robotic and electric devices have also been developed for training purposes in animal models to evaluate the functionality and effectiveness of the rehabilitation process. Similarly, forced exercise regimes, such as running on a treadmill or task-oriented motor training, are used to study rehabilitation scope in humans. Housing environments that provide social, motor and sensory stimuli and support cell engraftment, creating a more realistic approximation of human treatment, can be tested using animal models [103,104,105].

*Animal models in biomaterial testing*: Animal models have been well characterized for the study of brain tissues via brain atlases (http://www.med.harvard.edu/AANLIB/, https://portal.brain-map.org) for the required species. Stereotaxic techniques are utilized to introduce biomaterials or cells into particular coordinates of the target tissue. Microlesions can be studied precisely, and targeted localization can be confirmed using magnetic resonance imaging (MRI)-based lesion cartography [106,107,108].

## 6. Prevention and Treatment Strategies for Stroke

Stroke prevention involves modifying risk factors within a population or individuals, while stroke management depends on treating its pathophysiology. Despite an enormous amount of research into stroke over the last two decades, no simple means of treating or preventing all the clinical causes of stroke has been established. The overall direction of current stroke research is to generate novel therapies that modulate factors leading to primary and secondary stroke. Recent and current strategies for stroke prevention and treatment are discussed below (Figure 3).

*Excitotoxicity*: Neuronal death is a key manifestation of stroke. A key reason for this phenomenon is neuronal depolarization and inability to maintain membrane potential within the cell. This process is mediated by glutamate receptors N-methyl-D-aspartate (NMDA) and α-amino-3-hydroxy-5-methyl-4-isoxazolepropionic acid (AMPA), which were among the first neuroprotective agents tested in stroke prevention. However, the untimely release of glutamate overpowers the system that removes glutamate from the cell and causes abnormal release of NMDA and AMPA molecules, leading to uninhibited calcium influx and protein damage. As a result, these agents have not been shown to reduce neuronal death in human subjects. Targeting the molecular pathways downstream of excitotoxicity signaling, rather than directly targeting glutamatergic signaling, might reduce the side effects of the process [109,110].

*Gamma aminobutyric acid (GABA) agonists*: Clomethiazole is a GABA agonist that has been tested for its ability to improve stroke symptoms in patients, but failed to reduce the toxicity induced by the glutamate receptor [111].

*Sodium (Na^+^) channel blockers*: Na^+^ channel blockers have been used as neuroprotective agents in various animal models of stroke. They prevent neuronal death and reduce white matter damage. Many voltage-gated Na^+^ channel blockers have been tested in clinical trials, but most have proved to be ineffective [112]. Mexiletine is a neuroprotectant and Na^+^ channel blocker that proved effective in grey and white matter ischemic stroke, though further evaluation is required to confirm its role [113]. Lubeluzole was shown to reduce mortality in stroke in initial clinical trials, but successive trials failed to reproduce similar outcomes. Similarly, sipatrigine is a Na^+^ and Ca^2+^channel blocker which failed in a Phase II clinical trial in stroke patients. Amiodarone was shown to aggravate brain injury due to defective transportation and accumulation of Na^+^ ions in the brain after stroke [114].

*Calcium (Ca^2+^) channel blockers*: Voltage-dependent Ca^2+^ ion channel blockers have been shown to decrease the ischemic insult in animal models of brain injury. The Ca^2+^ ion chelator DP-b99 proved efficient and safe in Phase I and II clinical trials when administered to stroke patients. Similarly, Phase II trials significantly improved clinical symptoms in stroke patients treated within 12 h of onset [115]. In another study, Ca^2+^ channel blockers reduced the risk of stroke by 13.5% in comparison to diuretics and β-blockers [116].

*Antioxidants*: Reactive oxygen species produced in the normal brain are balanced by antioxidants generated in a responsive mechanism. However, in the ischemic stroke model, excess production of free radicals and inactivation of detoxifying agents cause redox disequilibrium. This phenomenon leads to oxidative stress, followed by neuronal injury. Therefore, antioxidants are employed in treatment of acute stroke to inhibit or scavenge free radical production and degrade free radicals in the system. In one study, antioxidant AEOL 10,150 (manganese (III) meso-tetrakis (di-N-ethylimidazole) porphyrin) effectively regulated the gene expression profiles specific to inflammation and stress response to decrease the ischemic damage and reperfusion in stroke patients [117]. In another, deferoxamine was shown to regulate the expression of hypoxia-inducible factor-1, a transcriptional factor regulated by oxygen levels, which in turn switched on other genes like vascular endothelial growth factor and erythropoietin. This mechanism, studied in an animal stroke model, proved beneficial in reducing lesion size and improving sensorimotor capabilities [118,119]. Similarly, NXY-059 compound acts as a scavenger to eliminate free radicals and decrease neurological deficits. The Stroke-Acute-Ischemic-NXY-Treatment-I (SAINT) clinical trial showed the efficacy and safety of NXY-059, but SAINT II failed to reproduce the positive effect of this drug in stroke patients [120,121]. In another study, researchers employed intravenous injection of antioxidants directly into mice brains to understand the benefits of route of administration. This method reduced neurological defects, but had minimal influence on brain damage [122].

### 6.1. Reperfusion

*The intravenous thrombolytics (IVT)*: The IVT treatment paradigm was originally developed to treat coronary thrombolysis but was found to be effective in treating stroke patients. The efficiency of thrombolytic drugs depends on factors including the age of the clot, the specificity of the thrombolytic agent for fibrin and the presence and half-life of neutralizing antibodies [123]. The drugs used in IVT treatment aim to promote fibrinolysin formation, which catalyzes the dissolution of the clot blocking the cerebral vessel. The most effective IVT drug, recombinant tissue plasminogen activator (rt-PA, or alteplase), was developed from research conducted by the US National Institute of Neurological Disorders and Stroke (NINDS) [124]. However, European Cooperative Acute Stroke Study (ECASS and ECASS II) researchers were unable to reproduce NINDS’ results. Later, it was found that this drug was effective in reducing clot diameter in stroke patients within three hours of incidence. The Safe Implementation of Thrombolysis in Stroke Monitoring Study (SITS-MOST) confirmed the efficacy and safety of alteplase within the designated time frame [125]. Another category of thrombolytics, consisting of fibrin and non-fibrin drugs, is used for treatment of stroke symptoms. Fibrin activators like alteplase, reteplase and tenecteplase convert plasminogen to plasmin directly, whereas non-fibrin activators like the drugs streptokinase and staphylokinase do so indirectly [123].

*Intra-arterial thrombolysis (IAT)*: IAT is another approach designed to combat acute stroke. This treatment is most effective in the first six hours of onset of MCA occlusion, and requires experienced clinicians and angiographic techniques [115]. Prolyse in Acute Cerebral Thromboembolism II (PROACT II) and Middle Cerebral Artery Embolism Local Fibrinolytic Intervention (MELT) were randomized clinical trials (RCTs) undertaken to test the efficacy and safety of a recombinant pro-urokinase drug [126,127], but did not produce any data useful for stroke treatment. Thrombolytics and glycoprotein IIb/IIIa antagonists were combined in two small clinical trials; this approach was helpful in treating atherosclerotic occlusions but less effective for cardioembolism [128,129]. The Interventional Management of Stroke (IMS) III trial tested IVT and IAT together to assess the benefits of combining rapid administration of therapy (IVT) and a superior recanalization methodology for faster relief (IAT) [130]. The IMS III trial was fruitful with bridging therapy (combination of IVT and IAT) as compared to IVT alone. There was an increase of 69.6% in the recanalization rate using bridging therapy in stroke patients [131,132].

*Fibrinogen-depleting agents*: Research has found a strong correlation between high fibrinogen levels in stroke patients and poor diagnosis for clinical outcomes. Fibrinogen-depleting agents decrease blood plasma levels of fibrinogen, hence reduce blood thickness and increase blood flow. They also remove the blood clot in the artery and restore blood flow in the affected regions of the brain. However, although some RCTs of defibrinogen therapy identified beneficial effects of fibrinogen-depleting agents in stroke patients, others failed to show positive effects on clinical outcomes after stroke [133]. Moreover, some studies reported bleeding after treatment with defibrinogen agents. Ancrod is a defibrinogenating agent derived from snake venom that has been studied for its ability to treat ischemic stroke within three hours of onset [134]. The European Stroke Treatment with Ancrod Trial (ESTAT) concluded that controlled administration of ancrod at 70 mg/dL fibrinogen was efficacious and safe, and achieved lower prevalence of ICH than observed at lower fibrinogen levels [135].

### 6.2. Others

*Antihypertensive therapy*: Hypertension is a risk factor for stroke. There are many reasons for high BP in stroke, including a history of hypertension, acute neuroendocrine stimulation, increased intracranial pressure, stress linked to hospital admission and intermittent painful spells [136]. Correct treatment of high BP during stroke is uncertain due to contradictory outcomes of clinical studies. Some research shows positive correlations between high BP and stroke-related mortality, hematoma expansion or intracerebral damage, suggesting that high BP should be treated. In other studies, low BP levels led to tissue perfusion and increased lesion size, thereby worsening the clinical outcome [137,138]. The multi-center Acute Candesartan Cilexetil Therapy in Stroke Survivors (ACCESS) Phase II study proved that taking medication (candesartan) for BP during stroke was safe, with no orchestrated cerebrovascular events reported due to hypotension. Similar research has been performed with antihypertensive drugs, such as the Continue Or Stop post Stroke Antihypertensives Collaborative Study (COSSACS) to study the efficacy of antihypertensive therapy in stroke; the Control of Hypertension and Hypotension Immediately Post Stroke (CHHIPS) study, designed to determine the cut-off value for BP during an attack; and the Scandinavian Candesartan Acute Stroke Trial (SCAST), which aimed to measure the effectiveness of the drug candesartan on stroke and cardiovascular disease [115,139]. In the COSSACS study, continuing antihypertensive drugs for a two-week period produced no extra harm as compared to stopping it and might be associated with reduced two-week mortality in patients with ischemic stroke [140]. The CHHIPS study demonstrated that a relatively moderate reduction in blood pressure lowered the mortality rate [141], whereas the SCAST study suggested that a careful BP-lowering treatment was associated with a higher risk of poor clinical outcome [142]. 

*Glucose management*: Hyperglycemia (elevated blood glucose) is common in stroke patients, so targeting blood glucose levels is an efficient stroke management strategy. Hyperglycemia > 6.0 mmol/L (108 mg/dL) is observed in most stroke patients; it initiates lipid peroxidation and cell lysis in compromised tissue, leading to stroke complications. An experimental study conducted in a rat model of collagenase-induced ICH found that hyperglycemia worsens edema formation and increases cell death, accelerating the course of ischemic injury. Increased blood glucose level is also associated with progression of infarction, reduced recanalization and poor clinical outcome [143]. Continuous glucose monitoring systems have been deployed to reduce stroke-related risks in both diabetic and non-diabetic stroke patients [144].

*Antiplatelet therapy*: This therapy is used for acute ischemic stroke management and for prevention of stroke incidence. It is also vital in controlling non-cardioembolic ischemic stroke and TIA. Antiplatelet agents like aspirin, clopidogrel and ticagrelor are the most widely used drugs administered to stroke sufferers within the first few days of attack [145]. Dual antiplatelet therapy, which involves a combination of clopidogrel, prasugrel or ticagrelor with aspirin, has become popular; many studies have tested the efficacy and safety of this dual therapy. It has been claimed that clopidogrel and aspirin combination therapy is most beneficial if introduced within 24 h of stroke and continued for 4–12 weeks [146].

*Stem cell therapy*: It offers promising therapeutic opportunities, safety and efficacy to stroke patients. Research on embryonic stem cells, mesenchymal cells and induced pluripotent stem cells has assessed their potential for tissue regeneration, maintenance, migration and proliferation, rewiring of neural circuitry and physical and behavioral rejuvenation [147]. Recently, a new type of mesenchymal stem cells (MSCs), called multilineage differentiating stress-enduring (Muse) cells, has been found in connective tissue. These cells offer great regenerative capacity and have been tested as a stroke treatment. After intravenous transplantation of Muse cells in a mouse model, they were found to engraft into the damaged host tissue and differentiate to provide functional recovery in the host [148]. Neovascularization is another mode of action of cell therapies in stroke; studies conducted in vitro and in vivo have shown that transplanted cells promote angiogenesis [149,150]. Furthermore, multiple stroke studies have reported that MSCs stimulate neurogenesis; this was confirmed in human embryonic neural stem cells using BrdU-labelling [151,152]. Stem cell therapy enhances the proliferation of neural stem cells and neuritogenesis [153]. Careful experimental design and clinical trials of stem cell therapies are likely to usher in a new era of treatment for stroke by promoting neurogenesis, rebuilding neural networks and boosting axonal growth and synaptogenesis.

*Neural repair*: This is an alternative therapy to neuroprotection. It is used to rejuvenate the tissue when the damage is already done and is therefore not time-bound but is most effective when administered 24 h after stroke attack. Many animal models have been used in attempts to stimulate neurogenesis and initiate the neuronal repair process [154]. Neural repair utilizes stem cell therapy to initiate repair mechanisms through cell integration into the wound or use of neurotrophic factors to block neuronal growth inhibitors. These cells may be channeled to any injured region to facilitate greater synaptic connectivity. Clinical trials using neural stem cells have proven beneficial in stroke patients. However, trials of myelin-associated glycoprotein, neurite outgrowth inhibitor (NOGO) proteins and chondroitin sulphate proteoglycans have shown these agents to be insufficiently effective; more clinical trials are required to increase treatment efficacy [155]. Biological intrusions may foster regeneration of newer cells, improve axonal guidance and enhance neural circuitry. Pharmacological and immunological interventions may target receptors to provide signaling cues for regeneration or block inhibitory factors in stroke-affected regions of the brain [156].

*Rehabilitation*: Stroke can leave individuals with short- and long-term disabilities. Daily activities like walking and toileting are often affected, and sensorimotor and visual impairment are common. Rehabilitation aims to reinforce the functional independence of people affected by stroke [157]. It includes working with patients and families to provide supportive services and post-stroke guidance after 48 h of stroke attack in stable patients. Stroke rehabilitation may involve physical, occupational, speech and/or cognitive therapy. It is designed to assist patients to recover problem-solving skills, access social and psychological support, improve their mobility and achieve independent living. Rehabilitation may also include neurobiological tasks designed to lessen the impact of cognitive dysfunction and induce synaptic plasticity, as well as long-term potentiation [158,159]. Neuromodulators play a vital role in triggering expression of specific genes that promote axon regeneration, dendritic spine development, synapse formation and cell replacement therapy. Task-oriented approaches, like arm training and walking, help stroke patients to manage their physical disability, and visual computer-assisted gaming activities have been used to enhance visuomotor neuronal plasticity [160].

## 7. Trends in Stroke Research

The incidence of stroke-related emergencies has decreased substantially over recent years due to improved understanding of the pathophysiology of stroke and identification of new drugs designed to treat the multitude of possible targets. Technological advancements like telestroke [161] and mobile stroke [162] units have reduced mortality and morbidity. Therefore, stroke management systems should include post-stroke care facilities on top of existing primary care and access to occupational, speech or any physical therapy following hospital discharge. Hospitals should develop standardized policies to handle emergencies in a timely fashion to avoid casualties and prevent secondary stroke [163]. Recently, the role of physiotherapists has emerged as an important aspect of post-stroke care management. Physiotherapists have initiated clinical trials of stroke recovery processes and rehabilitation therapy sessions. One ongoing study includes a strategy to manage disability by improving mobility using treadmill exercise, electromechanical device therapy and circuit class therapy [164,165]. Stroke Recovery and Rehabilitation Roundtables bring physiotherapists and other experts together to recommend research directions and produce guidance for the post-stroke healthcare system. Optimized delivery of stroke care systems and access to rehabilitation services are the future of healthcare for stroke [166].

Animal models used in stroke research reflect only a portion of the consequences of the condition in human subjects. Moreover, experiments conducted within a single laboratory are often constrained in terms of their research output. In vivo animal models of stroke should include aged populations to maximize their relevance, but most recent studies involve young and adult animals. Stroke studies should be conducted in both male and female subjects to exclude gender bias, and should take account of other confounders like hypertension, diabetes and obesity. All these issues make stroke research complex and expensive, and imply that it should be carried out collaboratively, across multiple labs. Ideally, an international multicenter platform for clinical trials would be established to increase the validity of research outcomes with respect to efficacy, safety, translational value, dose–response relationships and proof-of-principle. This strategy will help to overcome the current hurdles in transforming laboratory data into therapeutics for stroke.

Advancements in stem cell technologies and genomics have led to regenerative therapy to rebuild neural networks and repair damaged neurons due to ischemic insult [167,168]. The WIP1 gene is a regulator of Wnt signaling and a promising target for drug development. Studies in mice models showed that knockdown of WIP1 downregulates the stroke functional recovery process after injury, and that the presence of this gene regulates neurogenesis through activation of β-Catenin/Wnt signaling [169]. Similarly, NB-3 (contactin-6) plays a vital role in neuroprotection, as shown by knockdown of NB-3 in mice after stroke attack. NB-3-deficient mice had increased brain damage after MCAo, which also affected neurite outgrowth and neuronal survival rate. NB-3 is believed to have therapeutic benefits for ischemic insult [170]. Therefore, WIP1 and NB-3 are promising candidates for future drug trials. This is a vast field, and more research must be conducted in the coming years to enable the development of therapeutic drugs.

Numerous natural compounds have proven to be beneficial for stroke prevention and treatment. They can be synthesized at a lower cost than synthetic compounds and offer competitive efficacy and safety. Honokiol is a natural product that showed neuroprotective effects in animal models, and appears to have a role in reducing oxidative stress and inhibiting inflammatory responses [171]. Gastrodin, a compound extracted from *Gastrodia elata*, is a promising candidate in stroke treatment. In a mouse model, it improved neurogenesis and activated β-Catenin-dependent Wnt signaling to provide neuroprotection after ischemic insult. It also has antioxidative effects which protects the neural progenitor cells from neuron functional impairment. Gastrodin’s safety has been proved in clinical trials, hence it is an option for stroke management in the coming years [172].

The Utstein methodology is a process of standardizing and reporting research on out-of-hospital stroke and defining the essential elements of management tools. Its growing popularity led to the establishment of the Global Resuscitation Alliance (GRA), an organization that governs best practices. The primary aim of GRA is to facilitate stroke care from pre-hospital admission to rehabilitation and recovery. It has developed 10 guidelines to ensure smooth transitioning of services during and after attack. It has implemented a stroke registry, public awareness and educational programs, promoted techniques for early stroke recognition by first responders, sought to optimize prehospital and in-hospital stroke care, advocated the use of advanced neuroimaging techniques and promoted a culture of excellence. The Utstein community has developed comprehensive plans to improve early diagnosis and treatment of stroke patients globally [173].

Future clinical trials should aim not only to determine the efficacy and safety of drugs but to characterize recovery and clinical outcomes. Clinical trials of pharmacological therapies for post-stroke recovery should adhere to the following guidelines [174]. Patients should be enrolled within two weeks of stroke whenever possible. Studies should include sampling from a multicenter platform and include global scale criteria for data analysis. The underlying mechanism of action of the tested drugs on target molecules should be thoroughly understood. Secondary measurements like day-to-day progress of recovery, length of rehabilitation, treatment endpoint analysis and any other compounding factors should also be recorded. Overall, research on stoke management has advanced rapidly in recent years and is certain to make additional valuable discoveries through the application of new technologies in hypothesis-driven clinical trials.

## 8. Translational Challenges for the Current Stroke Therapeutic Strategies

Stroke research has seen fundamental advancements over recent years. The improvements in the selection of animal models, imaging techniques and methodological progress have led to immense drug targets and therapeutic interventions. In spite of this, the subsequent clinical trials failed to prove pre-clinical outcomes. Recanalization therapy showed some promising results in the clinical trials but only a small section of stroke patients benefited from this treatment [175]. Hence, the translational potential of stroke research is still under-investigated.

The key challenges that hinder the smooth transition of pre-clinical research into successful drugs include relevant endpoint selection, confounding diseases models like hypertension and diabetes, modelling age and gender effects in stroke patients, development of medical devices, investigating medical conditions that co-exist during stroke incidence, reproducibility of pre-clinical stroke research data and modelling functional and behavioral outcome [176,177,178]. Multiple causality of the stroke occurrence is another problem that is often over-looked. Homogeneity in stroke models to exhibit the broad spectrum of stroke pathophysiology associated with ischemic lesions or cortical or intracerebral damage is critical. Therefore, stroke animal models that target specific causes of stroke should be included. Latent interaction between comorbidities and stroke treatment should be identified to increase the safety and efficacy of the clinical outcome [179]. Short-term experimental trials often result in failed therapeutic development due to false-negative outcomes in the clinical settings [180]. Understanding the functional and behavioral output which might mislead true recovery is problematic in clinical trials wherein animal models have greater ability to mask the functional benefits [181]. This affects the affecting translational capability of the research. Adapting a combined approach to model recovery and rehabilitation is also important for successful transition.

One of the other problems with the clinical trials for stroke is the lack of efficient data management. The impact of large data generated from numerous clinical experiments is over-whelming and there should be a standardized system to manage such data. Moreover, these data should be deposited into a public data repository for easy access.

Industry and academic corroborations in stroke research are critical to improve the translational value [182]. A consensus between industry and academic interests is vital for successful transition. The industry collaborations are mostly monetary driven and have time constraints which might compromise the pre-clinical study protocol design, appropriate sample sizes and overestimation of treatment effects. IP protection and publication of research data may discord between these groups. A multicenter approach, long-term collaborations, effective project management, use of advanced methodologies and establishment of functional endpoints will probably advance the translational roadblocks in stroke research [183].

## 9. Conclusions

Stroke is the second leading cause of death and contributor to disability worldwide and has significant economic costs. Thus, more effective therapeutic interventions and improved post-stroke management are global health priorities. The last 25 years of stroke research has brought considerable progress with respect to animal experimental models, therapeutic drugs, clinical trials and post-stroke rehabilitation studies, but large gaps of knowledge about stroke treatment remain. Despite our increased understanding of stroke pathophysiology and the large number of studies targeting multiple pathways causing stroke, the inability to translate research into clinical settings has significantly hampered advances in stroke research. Most research has focused on restoring blood flow to the brain and minimizing neuronal deficits after ischemic insult. The major challenges for stroke investigators are to characterize the key mechanisms underlying therapies, generate reproducible data, perform multicenter pre-clinical trials and increase the translational value of their data before proceeding to clinical studies.

## Figures and Tables

**Figure 1 ijms-21-07609-f001:**
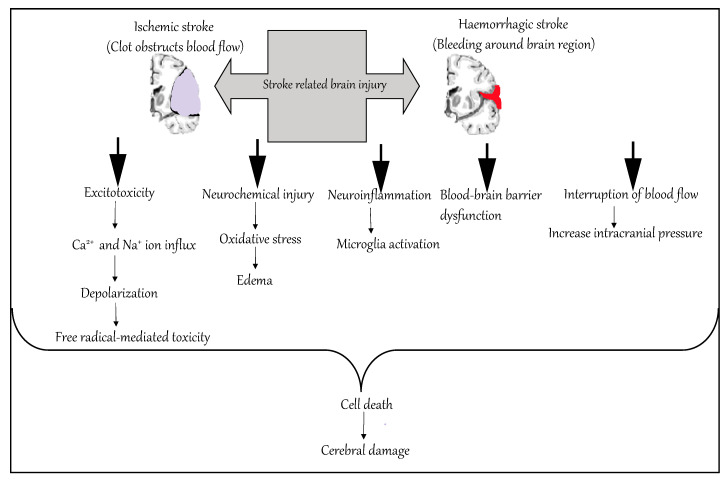
Molecular mechanism of stroke.

**Figure 2 ijms-21-07609-f002:**
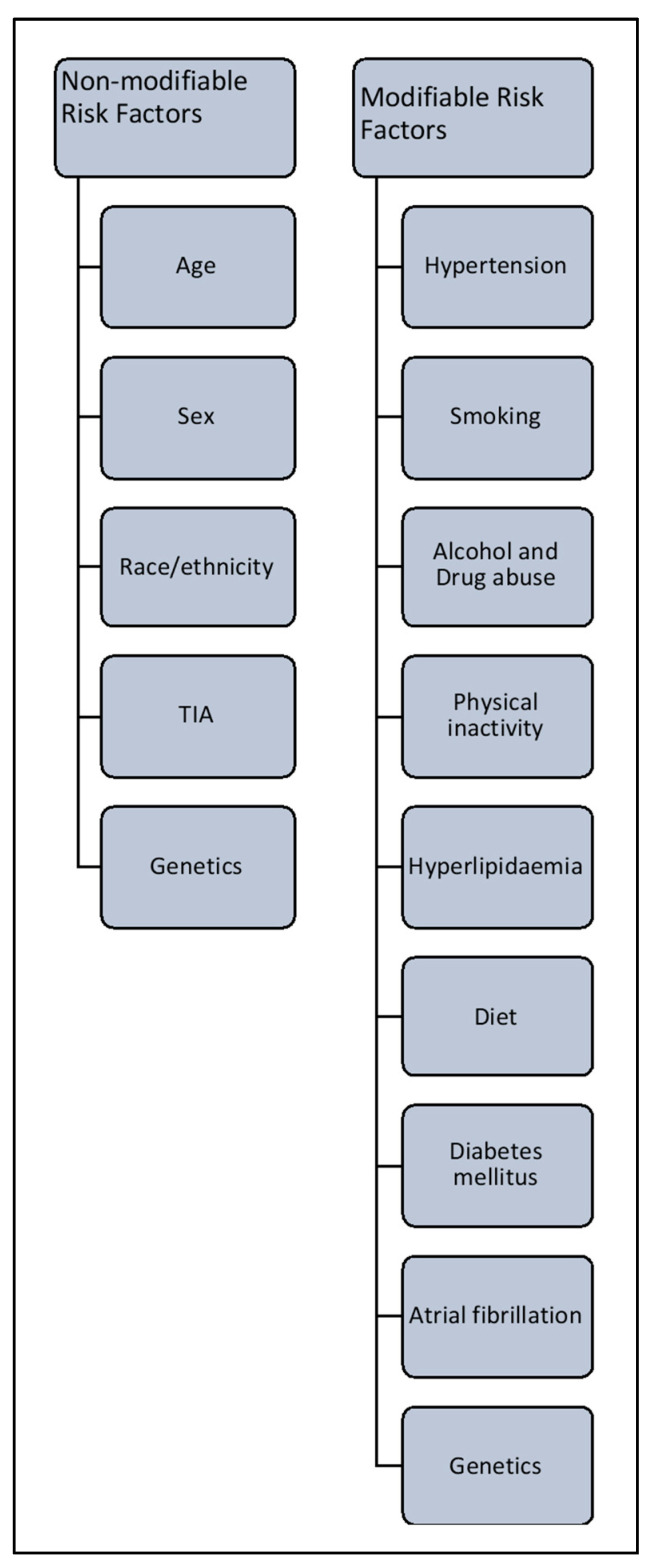
Risk factors associated with stroke.

**Figure 3 ijms-21-07609-f003:**
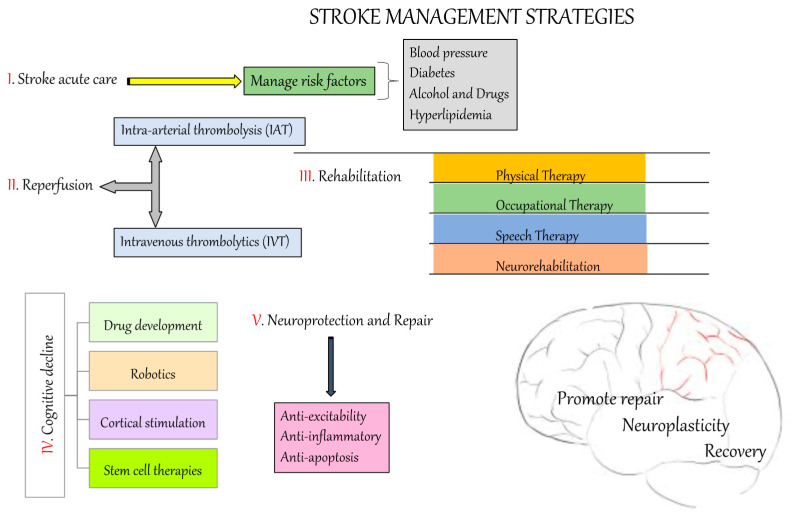
Stroke therapy. This represents the overall process to manage the incidence of stroke.

**Table 1 ijms-21-07609-t001:** Advantages and disadvantages of the stroke models.

Stroke Models	Advantages	Disadvantages
Intraluminal suture MCAo model	Mimics human ischemic stroke, Exhibits a penumbra, Highly reproducible, No craniectomy	Hyper-/hypothermia, Increased haemorrhage, Not suitable for thrombolysis studies
Craniotomy model	High long-term survival rates, Visual confirmation of successful MCAo	Highly invasive and procedural complications, Requires surgical skills
Photo-thrombosis model	Enables well-defined localization of an ischemic lesion, Highly reproducible, Less invasive	Causes early vasogenic edema, Not suitable for investigating neuroprotective agents
Endothelin-1 model	Less invasive, Induction of ischemic lesion in cortical regions, Low mortality	Duration of ischemia not controllable, Induction of astrocytosis and axonal sprouting
Embolic stroke model	Mimics the pathogenesis of human stroke	Low reproducibility of infarcts, Spontaneous recanalization
Neurorehabilitation	Rapid establishment of independence in activities of daily living Improves outcomes for cognitive, language, and motor skills	Develop cost-effective rehabilitative services Lack of in-depth studies on efficacy related to neurorehabilitation
Biomaterial testing	Reduction in lesion volume Bridge the lesion with neural tissue for neural reorganization Reduce secondary damage Improve neurological behaviour	Long-term experiments with the same biomaterial are challenging because of the degradation of material which might affect the treatment

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
