# Peer review of "Pathophysiology and Treatment of Stroke: Present Status and Future Perspectives"

_ijms, 2020, doi:10.3390/ijms21207609_

Round 1

Reviewer 1 Report

The manuscript of Kuriakose and Xiao entitled “Pathophysiology and Treatment of Stroke: Present Status and Future Perspectives” reviews epidemiology, pathophysiology, risk factors, animal models, prevention and treatment strategies, and trends in stroke (research) in a very elegant and well-arranged manner. The manuscript gives an excellent overview on the human disease and combines it with results and prospects obtained in animal models.

Minor comments:

  • Please add references to the paragraph “age-specific stroke” line 54, “gender-specific stroke” line 58 and lines 98-100.
  • The figure 1 could be colored comparable as figure 2.

Author Response

Comment#1: Please add references to the paragraph “age-specific stroke” line 54, “gender-specific stroke” line 58 and lines 98-100.

Response: Reference added in word document.

Comment#2: The figure 1 could be coloured comparable as figure 2.

Response: Figure Coloured

Word document attached herewith

Reviewer 2 Report

This is a well written review paper. The authors comprehensively discussed the pathophysiology, risk factors, current treatment strategies and future trend for stroke therapy and research. In addition, this paper discussed stroke treatment from both clinical and animal research. This review paper will be of great readership for both basic neuroscientists and medical professionals. I have some minor suggestions for the authors: 

  1. A diagram summarizing the molecular mechanisms following stroke brain injury is necessary, and will make this paper more strong.
  2. What are the translational challenges for the current stroke therapeutic strategies? Please discuss in a separate section.
